# Comparative Efficacy and Safety of Programmed Death-1 Pathway Inhibitors in Advanced Gastroesophageal Cancers: A Systematic Review and Network Meta-Analysis of Phase III Clinical Trials

**DOI:** 10.3390/cancers13112614

**Published:** 2021-05-26

**Authors:** Laercio Lopes da Silva, Pedro Nazareth Aguiar, Robin Park, Eduardo Edelman Saul, Benjamin Haaland, Gilberto de Lima Lopes

**Affiliations:** 1Metrowest Medical Center, Department of Medicine, Tufts University School of Medicine, Framingham, MA 02111, USA; Robin.Park@mwmc.com; 2Faculdade de Medicina do ABC, Instituto Oncoclinicas, Av. Juscelino Kubitschek, 510, São Paulo 04543-906, SP, Brazil; pedro.aguiar@prestadores.amil.com.br; 3Department of Medicine, Jackson Memorial Hospital, University of Miami, Miami, FL 33136, USA; eduardo.saul@jhsmiami.org; 4Population Health Sciences and Huntsman Cancer Institute, University of Utah, Salt Lake City, UT 84112, USA; ben.haaland@hsc.utah.edu; 5Division of Hematology-Oncology, Sylvester Comprehensive Cancer Center, University of Miami, Miami, FL 33146, USA

**Keywords:** immunotherapy, gastric cancer, esophageal cancer, multimodality treatments, clinical trials, meta-analysis

## Abstract

**Simple Summary:**

The use of checkpoint inhibitors has changed the treatment landscape for gastroesophageal cancer in the third-line setting. However, success rates in earlier treatment lines are highly variable across trials. Herein, we compare the efficacy and safety of the different anti-PD-1/PD-L1 regimens with or without chemotherapy.

**Abstract:**

*Background*: The use of checkpoint inhibitors has changed the treatment landscape for gastroesophageal cancer in the third-line setting. However, success rates in earlier treatment lines are highly variable across trials. Herein, we compare the efficacy and safety of the different anti-PD-1/PD-L1 regimens with or without chemotherapy; *Methods*: We performed a network meta-analysis (NMA) of anti-PD-1/PD-L1 monotherapy or combined with chemotherapy (chemoimmunotherapy) for gastroesophageal cancers without ERBB2 overexpression; *Results*: The first-line NMA included four trials (N = 3817), showing that chemoimmunotherapy improved OS and PFS without significant safety difference: Nivolumab-chemotherapy, OS (HR: 0.83 [95% CI, 0.75–0.92]), PFS (HR 0.68 [95% CI, 0.57–0.81]), Pembrolizumab-chemotherapy: OS (HR 0.77 [95% CI, 0.67–0.88]), PFS (HR: 0.72 [95% CI, 0.60–0.85]. Pembrolizumab monotherapy was the safest first-line treatment, SAE (OR 0.02 [95% CI, 0.00–0.2]) but showed no survival benefit. The second-line NMA encompassed four trials (N = 2087), showing that anti-PD-1 significantly improved safety but not survival: camrelizumab, SAE (OR 0.37; [95% CI, 0.24–0.56]); nivolumab, SAE (OR 0.13, [95% CI, 0.08–0.2]) pembrolizumab, SAE (OR 0.4; [95% CI, 0.30–0.53]); *Conclusions*: chemoimmunotherapy improves OS and PFS in previously untreated gastroesophageal cancers. Anti-PD-1 monotherapies improve safety in refractory disease, with no significant survival benefit.

## 1. Introduction

Gastric and esophageal cancers are the third and sixth leading causes of cancer mortality worldwide, with an estimated 768,793 and 544,076 deaths in 2020, respectively [1]. The diagnosis usually occurs in patients with locally advanced unresectable or metastatic disease, when treatment options are limited and with no curative intent. Chemotherapy remains the primary way to improve survival and quality of life in patients with gastroesophageal cancer. For those without overexpression of ERBB2 (previously, HER2), the first-line treatment is usually a choice of a platinum-fluoropyrimidine doublet, resulting in median survival of one year [2,3,4,5]. In the second-line setting, a taxane (docetaxel, paclitaxel) or irinotecan can improve survival in patients with good performance status [6,7,8]. However, the median overall survival is only six months, with a more significant benefit in patients that progressed 3–6 months after first-line chemotherapy. [7] Ramucirumab, an anti-VEGFR, showed similar survival benefits to chemotherapy as a single agent [9], while improving overall survival from 5.9 to 8.5 months when combined with paclitaxel as a second-line treatment [10].

Immune checkpoint inhibitors (ICI) targeting the programmed death-1 (PD-1)/programmed cell death ligand-1 (PD-L1) pathways are more established treatment options for patients with gastroesophageal cancer that progressed after two or more chemotherapy lines [11,12]. However, immunotherapy had not significantly improved survival in earlier therapy lines until recently [13,14,15,16,17]. Preliminary results from the KEYNOTE-590 and CheckMate 649 presented at the 2020 European Society for Medical Oncology (ESMO) annual meeting showed combinations of anti-PD-1 drugs with chemotherapy (chemoimmunotherapy) might be more effective than chemotherapy alone [18,19].

This study aimed to compare the efficacy and safety of PD-1/PD-L1 inhibitors for patients with advanced ERBB2 negative gastric and esophageal cancers. We performed a comprehensive analysis of the current data published from phase III randomized clinical trials (RCT) to inform decision making and enable the development of optimal first- and second-line treatment strategies for those patients.

## 2. Materials and Methods

We performed our study under the extension for network meta-analysis from the Preferred Reporting Items for Systematic Reviews and Meta-analyses (PRISMA) [20,21]. We created a prospective protocol and uploaded it to PROSPERO (CRD42020221822).

### 2.1. Eligibility Criteria

We considered eligible all randomized clinical trials comparing PD-1/PD-L1 inhibitors or anti–PD-L1, as single agents or combined with chemotherapy versus chemotherapy alone, in patients with esophageal, gastric, and gastroesophageal junction tumors, in the frontline or second-line treatments. We considered ineligible trials in phases 1 or 2 and trials that compared PD-1/PD-L1 inhibitors with other immunotherapies. When we found multiple references for the same study, we favored the latest and most complete report.

### 2.2. Data Sources and Extraction

We performed an extensive database search (PubMed, Embase, Cochrane Central, Web of Science, Medline, Scopus, and ClinicalTrials.gov) for entries from 1 January 2010 to 23 November 2020. We also reviewed abstracts from the American Society of Clinical Oncology and the ESMO libraries until 21 November 2020. A detailed search strategy is available in Table A1.

We uploaded titles and abstracts to Rayyan QCRI, a web-based platform for systematic review management. [22] Three authors independently performed the screening. Data from the included trials was performed by two authors, in tandem, and using a pre-piloted spreadsheet containing trial identification, baseline patient characteristics (including PD-L1 expression status), treatments, and outcomes. We resolved discrepancies by consensus. The efficacy outcomes of interest were overall survival (OS) and progression-free survival (PFS). The safety outcome of interest was the incidence of serious adverse events (SAEs), characterized as treatment-related adverse events (TRAEs) grade 3 to 5.

### 2.3. Risk of Bias Assessment

We used the Cochrane Collaboration’s tool (version 2.0), which includes five domains (randomization process, deviation from intended interventions, missing outcome data, measurement of the outcome, and selection of reported results) and results in judgments of “low risk of bias”, “some concerns”, or “high risk of bias” [23]. Two authors independently applied the tool to each included trial. Any inconsistencies were solved by a discussion and between the authors.

### 2.4. Statistical Analysis

We performed a network meta-analysis with a frequentist approach and a random-effects model using the package ‘netmeta’ for R statistical software (version 4.0.3, R Project for Statistical Computing) [24]. We used multivariate normal distribution and random-effects models to account for between-arm correlation in multi-arm trials inside the frequentist network [25]. We generated forest plots for back-transformed network estimates. We assessed heterogeneity between and within designs using Cochran’s Q statistics and quantified using I2 statistics. I2 can be used to describe the proportion of the variability in effect estimates due to heterogeneity within three thresholds 25% (low), 50% (moderate) and, 75% (high) [26,27]. We expressed OS and PFS outcomes as hazard ratios (HR) with the respective 95% confidence interval (95% CI) and SAEs as odds ratios (OR) with the respective 95% CI.

## 3. Results

### 3.1. Study Selection

We found a total of 2386 unique entries. After excluding duplicates, we screened titles and abstracts for 1000 records. We assessed 149 full-text publications, including 12 trial registrations (Figure A1). We included eight trials in the quantitative synthesis: four in the first-line setting and four in the second-line setting.

### 3.2. First-Line Treatments

#### 3.2.1. Study Characteristics

The four trials in the first-line setting involved 3817 patients. ATTRACTION-4 and KEYNOTE-649 evaluated nivolumab + chemotherapy (Nivo-Chemo). KEYNOTE-590 evaluated Pembrolizumab + chemotherapy (Pembro-Chemo). KEYNOTE-062 had three-arms, comparing Pembrolizumab monotherapy (Pembro) or Pembro-Chemo with chemotherapy alone (Table 1). KEYNOTE-062 included only patients with PD-L1 combined positive score (CPS) ≥ 1. Further details PD-L1 expression subgroups in each included trial can be found in the Table A2.

#### 3.2.2. Network Meta-Analysis

We found that the combination of anti-PD-1 with chemotherapy improves survival. Pembro-Chemo showed similar OS benefit (HR, 0.77; 95% CI, 0.67–0.88) to Nivo-Chemo (HR, 0.83; 95% CI, 0.75–0.92). PFS was also comparable with Nivo-Chemo (HR, 0.68; 95% CI, 0.57–0.81) and Pembro-Chemo (HR, 0.72; 95% CI, 0.60–0.85). Pembro monotherapy did not improve survival, OS (HR, 0.91; 95% CI, 0.71–1.16), PFS (HR, 1.62; 95% CI, 1.23–2.14) but showed a markedly better safety profile than chemotherapy, SAE (OR, 0.02; 95% CI, 0.00–0.2). We found no significant safety difference in SAE from Nivo-Chemo (OR 0.54; 0; 95% CI, 0.1–2.92) or Pembro-Chemo SAE (OR, 1.31; 0; 95% CI, 0.23–7.35), compared with chemotherapy (Figure 1A).

#### 3.2.3. Risk of Bias

All four trials had a low risk of bias for OS. We considered CheckMate 649 a high risk of bias for PFS and SAEs, mostly related to missing outcome data. The remaining studies had a low risk of bias for PFS and SAEs (Table 2).

### 3.3. Second-Line Treatments

#### 3.3.1. Study Characteristics

The studies in the second-line setting involved 2087 individuals. All trials (KEYNOTE-061, ATTRACTION-3, KEYNOTE-181, and ESCORT) compared chemotherapy with pembrolizumab, nivolumab, or camrelilzumab. The predominant tumor site was esophageal (Table 3). All studies had subgroups according to PD-L1. All publications had OS, PFS, and SAEs data available (Table A3).

#### 3.3.2. Network Meta-Analysis

Camrelizumab showed a greater survival benefit compared to chemotherapy, OS (HR 0.71; 95% CI, 0.54–0.93), PFS (HR 0.69; 95% CI, 0.45–1.06); followed by nivolumab, OS (HR 0.77; 95% CI, 0.58–1.02), PFS (HR 1.08; 95% CI, 0.77–1.66) and pembrolizumab, OS (HR 0.86; 95% CI, 0.72–1.04), PFS (HR 1.28; 95% CI, 0.96–1.71). Anti-PD-1 drugs significantly improved safety, nivolumab had the lowest chance of serious adverse events (OR, 0.13; 95% CI, 0.08–0.2), followed by camrelizumab (OR, 0.37; 95% CI, 0.24–0.56) and pembrolizumab (OR, 0.4; 95% CI, 0.30–0.53) (Figure 1B).

#### 3.3.3. Risk of Bias 

The four trials had a low risk of bias for OS and PFS. For SAE, KEYNOTE-061 raised some concerns due to missing outcome data. The remaining studies had a low risk of bias for PFS (Table 4).

### 3.4. Subgroup Analysis: PD-L1 Expression

Published data from PD-L1 expression subgroups across trials was not consistent. There were variable cut-off values, so it was not statistically meaningful to add those subgroups in our network meta-analysis. To evaluate PD-L1 expression as a predictor of response to ICIs, we pooled the available OS HR from chemoimmunotherapy subgroups according to PD-L1 CPS. In the first-line setting, patients that overexpress PD-L1 had better OS, CPS ≥ 10 (HR 0.75; 95% CI, 0.65–0.87), CPS ≥ 5 (HR 0.71; 95% CI, 0.61–0.83); those with CPS ≥ 1, OS (HR 0.79; 95% CI, 0.71–0.89) had similar response to all randomized patients, OS (HR 0.80; 95% CI, 0.72–0.89) (Figure 2). In the second-line setting, we included all the single-agent PD-1 inhibitors: patients with PD-L1 CPS ≥ 1 had better OS (HR 0.71; 95% CI, 0.71–0.87) in comparison to all patients, OS (HR 0.83; 95% CI, 0.74–0.94), but patients with PD-L1 CPS ≥ 10 had a significant difference in OS (HR 1.19; 95% CI, 0.71–0.87) (Figure 3).

## 4. Discussion

The present study offers valuable insight on recent advances involving the use of PD-1 inhibitors in patients with advanced gastroesophageal cancers that do not overexpress ERBB2. For previously untreated patients, chemoimmunotherapy was the best strategy. Both Nivo-Chemo and Pembro-Chemo showed significantly better OS and PFS with no significant difference in SAEs. Conversely, pembrolizumab monotherapy was markedly safer than ICI-Chemo but did not improve OS and had the worst PFS.

For patients that progressed after one line of chemotherapy, our final selection encompassed three different anti-PD-1 drugs. Camrelizumab showed the best OS, followed by nivolumab and pembrolizumab. We did not observe the same benefit for PFS. We found that camrelizumab might improve PFS, but nivolumab and pembrolizumab might worsen PFS compared to chemotherapy. Importantly, PD-1 inhibitors were significantly safer than chemotherapy as second-line treatments. Nivolumab likely has the best profile, followed by camrelizumab and pembrolizumab.

A growing body of evidence shows that drugs such as cisplatin, oxaliplatin, and paclitaxel can up-regulate PD-L1 expression in tumor and immune cells, therefore blocking the chemotherapy effectiveness but opening an opportunity to the use of PD-1/PD-L1 inhibitors [29,30,31]. Other studies point that cytotoxic therapies can turn ‘cold’ tumors into ‘hot’ tumors by making them abundantly infiltrated by CD8+ T cells and dendritic cells, making them more susceptible to ICIs [32,33,34]. The impact of these changes in the tumor microenvironment in clinical effectiveness is yet to be proven [35]. However, they help explain why chemoimmunotherapy led to better survival outcomes than pembrolizumab alone in previously untreated patients, ref. [16] while single-agent PD-1 inhibitors provided better OS benefit in patients that progressed after chemotherapy [11,12].

The use of chemoimmunotherapy for a shorter period, followed by treatment with immunotherapy only, can lead to earlier disease control with more extended survival benefits and lower SAE rates. The CheckMate 9LA has recently demonstrated the benefit of such a strategy in patients with lung cancer [31]. Ongoing phase II trials such as “Blinded for peer review” (nivolumab with or without ipilimumab) and “Blinded for peer review” (avelumab) will help to identify optimal dosing and administration schedules of immunogenic chemotherapy for gastroesophageal cancers.

Several trials in our analysis did not achieve their primary endpoints, which can be related to heterogeneity inside the cohorts [13,14,15,16]. For instance, in KEYNOTE-062, all patients had CPS ≥1, and neither Pembro alone nor Pembro-Chemo significantly improved survival. In the subset of patients with CPS ≥ 10, Pembro prolonged OS (median 17.4 months versus 10.8 months; HR 0.69; 95% CI, 0.49–0.97), however no statistical test was applied to this difference [16]. Hence, identifying which tumors will respond to immune checkpoint inhibitors is paramount. Our analysis shows that PD-L1 CPS was not a robust predictor of efficacy, as the OS benefit from chemoimmunotherapy was similar OS across subgroups, which can be related to inconsistencies in PD-L1 assessment methods and cutoff values. Hopefully, the final results from ATTRACTION-4 and the ongoing KEYNOTE-859 will help consolidate the role of PD-L1 CPS in selecting patients for chemoimmunotherapy in the first-line setting. However, there is a need for alternative biomarkers.

## 5. Limitations

Our study’s first limitation comes to its nature as a network meta-analysis where we derived most of our conclusions from indirect comparisons. We used trial-level data rather than patient-level data, which could lower the power of our analysis.

Second, the trials had several differences in baseline characteristics that could affect the generalizability of the results. In the first-line, gastric and gastroesophageal adenocarcinomas were the predominant type, while in the second-line, esophageal (adenocarcinoma and squamous cell carcinoma) were the most frequent.

Third, the PFS and SAE data from CheckMate 649 included in our analysis in the first-line setting refers only to the subgroup of patients with PD-L1 CPS ≥ 5, which raises concerns for publication bias. We obtained most of the data in the first line from conference abstracts, and hopefully, further peer-reviewed publications will provide more detailed data from all patients included in each study.

## 6. Conclusions

Chemoimmunotherapy is the best first-line treatment for HER2 negative, advanced gastro-esophageal cancers. Nivo-Chemo and Pembro-Chemo improved OS and PFS similarly. Pembro did not improve survival but was significantly less toxic and should be considered as a first-line option. In the second-line setting, anti-PD-1 drugs might prolong survival, but camrelizumab was the only one to improve OS significantly. All anti-PD-1 drugs were significantly less toxic than chemotherapy for patients with refractory disease. The association of higher levels of PD-L1 expression with better outcomes remains unclear and would be better assessed in further analyses.

## Figures and Tables

**Figure 1 cancers-13-02614-f001:**
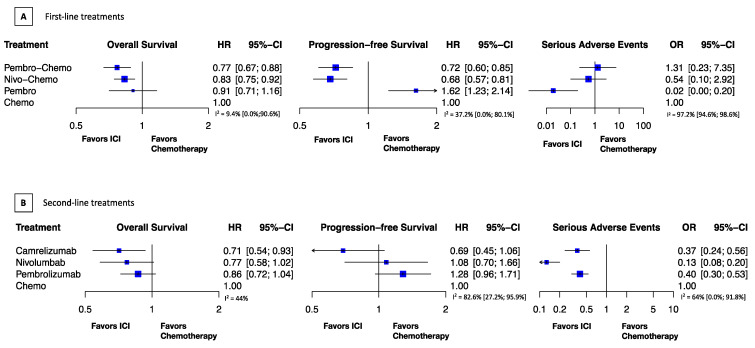
Relative treatment effects. HR indicates hazard ratio; (**A**). First-line treatments; (**B**). Second-line treatments; OR, odds ratio; CI, confidence interval. Treatment abbreviations: Chemo indicates chemotherapy; Pembro, pembrolizumab; Pembro-Chemo, pembrolizumab plus chemotherapy; Nivo-Chemo, nivolumab plus chemotherapy.

**Figure 2 cancers-13-02614-f002:**
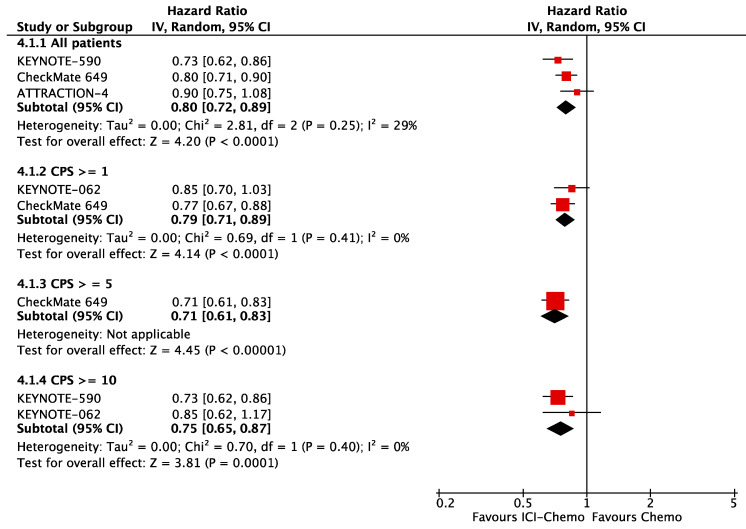
OS according to PD-L1 expression in the first-line setting.

**Figure 3 cancers-13-02614-f003:**
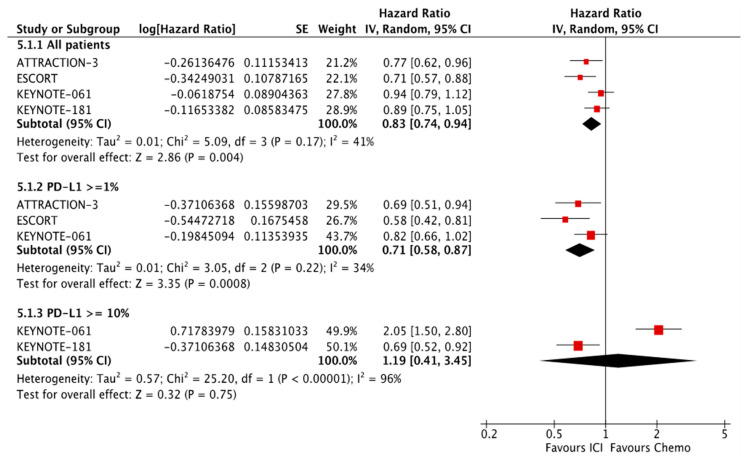
OS according to PD-L1 expression in the second-line setting.

**Table 1 cancers-13-02614-t001:** Characteristics of Trials Included in the Network Meta-Analysis of first-line treatments.

Study (Author, Year, [Reference])	Histology	Intervention (N)	Control (N)	Pertinent Characteristics
**ATTRACTION-4** **(Boku, 2020** **[28])**	NA	Nivo 360 mg Q3W + SOX Q3W or CapeOX Q2W (362)	SOX Q3W or CapeOX Q2W (362)	Asian: 100%
**KEYNOTE-062** **(Shitara, 2020** **[16])**	Adenocarcinoma (100%)	Pembrolizumab 200 mg Q3W (256) or Pembrolizumab 200 mg Q3W +cisplatin 80 mg/m^2^/d on day 1 + 5-FU 800 mg/m^2^/d on days 1–5 or capecitabine 1000 mg/m^2^ BID on days 1–14 every 3 weeks (257)	cisplatin 80 mg/m^2^/d on day 1+ 5-FU 800 mg/m^2^/d on days 1–5 or capecitabine 1000 mg/m^2^ BID on days 1–14 every 3 weeks (250)	Median age: 63.0 vs. 64.0 Male: 70% vs. 70%vs 75.9% vs. 71.6% Asian: 24.2% vs. 24.9% vs. 24.4%
**KEYNOTE-590** **(Kato, 2020** **[19])**	ESCC (73%) EGJ (27%)	Pembrolizumab 200 mg Q3W +chemo (cisplatin 80 mg/m^2^ Q3W [d1; 6 doses] + 5-FU 800 mg/m^2^ on d1–5 Q3W (373)	cisplatin 80 mg/m^2^ Q3W [d1; 6 doses] + 5-FU 800 mg/m^2^ on d1–5 Q3W (376)	Male: 83%
**CheckMate 649** **(Moehler, 2020** **[18])**	NA	nivolumab 360 mg Q3W or 240 mg Q2W (789)	XELOX Q3W or FOLFOX Q2W (792)	NA

**Table 2 cancers-13-02614-t002:** Risk of bias assessment, first-line trials.

Study ID	Experimental Arm	Outcome	Randomization Process	Deviations from Intended Interventions	Missing Outcome Data	Measurement of the Outcome	Selection of the Reported Result	Overall Bias
ATTRACTION-4	Nivo-Chemo	OS	Low	Low	Low	Low	Low	Low
KEYNOTE-590	Pembro-Chemo	OS	Low	Low	Low	Low	Low	Low
CheckMate 649	Nivo-Chemo	OS	Low	Low	Low	Low	Low	Low
KEYNOTE-062	arm 1 Pembro; arm2 Pembro-Chemo	OS	Low	Low	Low	Low	Low	Low
ATTRACTION-4	Nivo-Chemo	PFS	Low	Low	Low	Low	Low	Low
KEYNOTE-590	Pembro-Chemo	PFS	Low	Low	Low	Low	Low	Low
KEYNOTE-062	arm1 Pembro; arm2 Pembro-Chemo	PFS	Low	Low	Low	Low	Low	Low
CheckMate 649	Pembro-Chemo	PFS	Low	Some concerns	High	Some concerns	Low	High
ATTRACTION-4	Nivo-Chemo	SAE	Low	Low	Low	Low	Low	Low
KEYNOTE-590	Pembro-Chemo	SAE	Low	Low	Low	Low	Low	Low
KEYNOTE-062	arm 1 Pembro; arm 2 Pembro-Chemo	SAE	Low	Low	Low	Low	Low	Low
CheckMate 649	Nivo-Chemo	SAE	Low	High	High	High	Low	Low

**Table 3 cancers-13-02614-t003:** Characteristics of Trials Included in the Network Meta-Analysis of second-line treatments.

Study (Author, Year [Reference])	Histology	Intervention (N)	Control (N)	Pertinent Characteristics
**KEYNOTE-061** **(Shitara, 2019** **[13])**	Adenocarcinoma (79% vs. 79%) Tubular adenocarcinoma (7% vs. 10%)	Pembrolizumab 200 mg Q3W (296)	Paclitaxel 80 mg/m^2^ intravenously on days 1, 8, and 15 of 4-week cycles (296)	Median age: 62.5 vs. 60.0 Male: 68% vs. 70% Asian: 30% vs. 30%
**ATTRACTION-2** **(Kato, 2019** **[11])**	ESCC (100%)	Nivolumab 240 mg q2 weeks (each cycle was 6 weeks) (210)	Paclitaxel 100 mg/m^2^ q1 week for 6 weeks followed by 1 week off (each cycle was 7 weeks) and Docetaxel 75 mg/m^2^ q3 weeks (each cycle was 3 weeks) (209)	Median age: 64 vs. 67 Male: 85% vs. 89% Asian: 96% vs. 96%
**KEYNOTE-181** **(Kojima, 2020** **[15])**	ESCC (63.1% vs. 64.6%) EAC (36.9% vs. 35.4%)	Pembrolizumab 200 mg q3 weeks (314)	Paclitaxel 80–100 mg/m^2^ on days 1, 8, and 15 of each 28-day cycle, Docetaxel 75 mg/m^2^ on day 1 of each 21-day cycle, or irinotecan 180 mg/m^2^ on day 1 of each 14-day cycle (314)	Median age: 63 vs. 62 Male: 86.9% vs. 86.3 Asian: 38.5% vs. 38.9%
**ESCORT** **(Huang, 2020** **[17])**	ESCC (100%)	Camrelizumab 200 mg on day 1 of each 2-week cycle (228)	Docetaxel (75 mg/m^2^, on day 1 of each 3-week cycle) or irinotecan 180 mg/m^2^, on day 1 of each 2-week cycle) (220)	Median age: 60 vs. 60 Male: 91% vs. 87% Asian: 100%

**Table 4 cancers-13-02614-t004:** Risk of bias assessment, second-line trials.

Study ID	Experimental	Outcome	Randomization Process	Deviations from Intended Interventions	Missing Outcome Data	Measurement of the Outcome	Selection of the Reported Result	Overall Bias
KEYNOTE-061	Pembrolizumab	OS	Low	Low	Low	Low	Low	Low
ATTRACTION-3	Nivolumab	OS	Low	Low	Low	Low	Low	Low
ESCORT	Camrelizumab	OS	Low	Low	Low	Low	Low	Low
KEYNOTE-181	Pembrolizumab	OS	Low	Low	Low	Low	Low	Low
KEYNOTE-061	Pembrolizumab	PFS	Low	Low	Low	Low	Low	Low
ATTRACTION-3	Nivolumab	PFS	Low	Low	Low	Low	Low	Low
ESCORT	Camrelizumab	PFS	Low	Low	Low	Low	Low	Low
KEYNOTE-181	Pembrolizumab	PFS	Low	Low	Low	Low	Low	Low
KEYNOTE-061	Pembrolizumab	SAE	Low	Low	Some concerns	Low	Low	Some concerns
ATTRACTION-3	Nivolumab	SAE	Low	Low	Low	Low	Low	Low
ESCORT	Camrelizumab	SAE	Low	Low	Low	Low	Low	Low
KEYNOTE-181	Pembrolizumab	SAE	Low	Low	Low	Low	Low	Low

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
