# Peer review of "Comparative Efficacy and Safety of Programmed Death-1 Pathway Inhibitors in Advanced Gastroesophageal Cancers: A Systematic Review and Network Meta-Analysis of Phase III Clinical Trials"

_cancers, 2021, doi:10.3390/cancers13112614_

Round 1

Reviewer 1 Report

This manuscript entitled “Comparative Efficacy and Safety of Programmed Death-1 Pathway Inhibitors in Advanced Gastroesophageal Cancers: A Systematic Review and Network Meta-analysis of Phase III Clinical Trials.” by Laercio Lopes da Silva comprehensively review the current evidence of ICIs phase III studies in GE cancer and provide the reader a good sight for ICIs in GE cancer.

The most important issue is that not all the trials achieved pre-set statistical significance even the p valve is less than 0.05. Therefore, the study did not achieve the primary endpoints and did not have the indication from FDA. The authors should mention this point somewhere in the discussion to avoid misunderstanding for the readers.

Some other issues should be corrected or clarified before publication.

Abstract:

Line 25, “Four first-line trials (N =3817)” is not a sentence.

Line 29, Four second-line trials (N =2087) is not a sentence.

The format of conclusion is wrong.

Introduction

  1. “The introduction should briefly place the study in a broad context and highlight why 39 it is important. It should define the purpose of the work and its significance. The current 40 state of the research field should be carefully reviewed and key publications cited. Please 41 highlight controversial and diverging hypotheses when necessary. Finally, briefly men-42 tion the main aim of the work and highlight the principal conclusions. As far as possible, 43 please keep the introduction comprehensible to scientists outside your particular field of 44 References should be numbered in order of appearance and indicated by a nu-45 meral or numerals in square brackets—e.g., [1] or [2,3], or [4–6]. See the end of the docu-46 ment for further details on references.” should be deleted.
  2. Chemotherapy remains the primary way to improve survival and quality of life in patients with ERBB2 (previously HER2) negative tumors. The authors should mention HER2+ GC. In fact, chemotherapy is also the primary way for HER2+ GC as HER2-GC.
  3. Line 53, We usually use” median survival” rather than “average survival”
  4. Line 59, “to 8.5 months” authors should mention from xx to 8.5 months based on study.
  5. Line 69-70, HER2+ was not excluded in some trials. Did authors only analyze the subgroups of patients with HER2- patients?

3.2.1. Study characteristics

It is better to include the names of four trials in the section as shown in tables 1 & 2.

Table 1. include two KN-590, please check this.

The histology is lacking for Nivo trials.

Abbreviations such as XELOX, FOLFOX

Table 3. The abbreviations such as PTX, DTX

The format of citation is wrong in the discussion. Please correct this.

Author Response

We would like to thank the reviewers for their thoughtful comments and efforts towards improving our manuscript. In the following, we address comments specific to each reviewer:

Reviewer 2

 Abstract:

Line 25, “Four first-line trials (N =3817)” is not a sentence.

Line 29, Four second-line trials (N =2087) is not a sentence.

The format of conclusion is wrong.

Thank you for noticing these pertinent points, we adjusted the abstract accordingly:

“The first-line NMA included four trials (N =3817), showing that chemoimmunotherapy im-proved OS and PFS without significant safety difference: Nivolumab-chemotherapy, OS (HR: 0.83 [95% CI, 0.75-0.92]), PFS (HR 0.68 [95% CI, 0.57-0.81]), Pembrolizumab-chemotherapy: OS (HR 0.77 [95% CI, 0.67-0.88]), PFS (HR: 0.72 [95% CI, 0.60-0.85]. Pembrolizumab monotherapy was the safest first-line treatment, SAE (OR 0.02 [0.00-0.2]) but showed no survival benefit. The sec-ond-line NMA encompassed four trials (N =2087), showing that anti-PD-1 significantly im-proved safety but not survival: camrelizumab, SAE (OR 0.37; 95% CI [0.24-0.56]); nivolumab, SAE (OR 0.13, [0.08-0.2]) pembrolizumab, SAE (OR 0.4; [0.30-0.53]); Conclusions: chemoim-munotherapy improves OS and PFS in previously untreated gastroesophageal cancers. Anti-PD-1 monotherapies improve safety in refractory disease, with no significant survival benefit.”

Introduction

  1. “The introduction should briefly place the study in a broad context and highlight why 39 it is important. It should define the purpose of the work and its significance. The current 40 state of the research field should be carefully reviewed and key publications cited. Please 41 highlight controversial and diverging hypotheses when necessary. Finally, briefly men-42 tion the main aim of the work and highlight the principal conclusions. As far as possible, 43 please keep the introduction comprehensible to scientists outside your particular field of 44 References should be numbered in order of appearance and indicated by a nu-45 meral or numerals in square brackets—e.g., [1] or [2,3], or [4–6]. See the end of the docu-46 ment for further details on references.” should be deleted.

Thank you for noticing this formatting mistake. This paragraph was deleted from the manuscript.

  1. Chemotherapy remains the primary way to improve survival and quality of life in patients with ERBB2 (previously HER2) negative tumors. The authors should mention HER2+ GC. In fact, chemotherapy is also the primary way for HER2+ GC as HER2-GC.

This is an important point. We adjusted the sentence to “Chemotherapy remains the primary way to improve survival and quality of life in patients with gastroesophageal cancer”.

  1. Line 53, We usually use” median survival” rather than “average survival”

We replaced to expression to ‘median survival’, which is now in Line 52.

  1. Line 59, “to 8.5 months” authors should mention from xx to 8.5 months based on study.

This is an important point to understanding the problem presented. We improved the sentence as recommended, with “improving overall survival from 5.9 to 8.5 months” in Line 57.

  1. Line 69-70, HER2+ was not excluded in some trials. Did authors only analyze the subgroups of patients with HER2- patients?

Another important issue. Our analysis was focused on the patients without HER2 overexpression.

3.2.1. Study characteristics

It is better to include the names of four trials in the section as shown in tables 1 & 2.

We included the names in the study characteristics sessions

“The four trials in the first-line setting involved 3817 patients. ATTRACTION-4 and KEYNOTE-649 evaluated nivolumab + chemotherapy (Nivo-Chemo). KEYNOTE-590 evaluated Pembrolizumab + chemotherapy (Pembro-Chemo). KEYNOTE-062 had three-arms, comparing Pembrolizumab monotherapy (Pembro) or Pembro-Chemo with chemotherapy alone (Table 1)”…

“All trials (KEYNOTE-061, ATTRACTION-3, KEYNOTE-181, and ESCORT) compared chemotherapy with pembrolizumab, nivolumab, or camrelilzumab”.

Table 1. include two KN-590, please check this.

Thank you for noticing the mistake, we corrected the label to ‘ KEYNOTE-062’.

The histology is lacking for Nivo trials.

ATTRACTION-4 and CheckMate-649 were published in abstracts at ESMO 2020, and there was no information on the histology.

Abbreviations such as XELOX, FOLFOX

This is an important point. We kept these regimens as abbreviations due to limited space in the chart.

Table 3. The abbreviations such as PTX, DTX

We replaced the abbreviations to “Paclitaxel” and “Docetaxel”, respectively.

The format of citation is wrong in the discussion. Please correct this.

Thank you for noticing. We corrected the format.

Reviewer 2 Report

The use of checkpoint inhibitors has changed the treatment landscape for gastroesophageal cancer in the third-line setting. However, success rates in earlier treatment lines 21 are highly variable across trials. The authors performed a network meta-analysis of anti-PD-1/PD-L1 monotherapy or combined with chemotherapy (chemoimmunotherapy) using four first-line trials (N =3817) and four second-line trials (N =2087). They found that chemoimmunotherapy improves OS and PFS in previously untreated gastroesophageal cancers while anti-PD-1 monotherapies improve safety in refractory disease, with no significant survival benefit. Although the numbers of trials are small, the manuscript was well written and included their limitations. I have minor comments for this manuscript.  

In Introduction, please delete the first paragraph containing the recommendations from the editorial office. “The introduction should briefly place the study in a broad context and highlight why it is important. It should define the purpose of the work and its significance. The current state of the research field should be carefully reviewed and key publications cited. Please highlight controversial and diverging hypotheses when necessary. Finally, briefly mention the main aim of the work and highlight the principal conclusions. As far as possible, please keep the introduction comprehensible to scientists outside your particular field of research. References should be numbered in order of appearance and indicated by a numeral or numerals in square brackets—e.g., [1] or [2,3], or [4–6]. See the end of the document for further details on references.”

Author Response

We would like to thank the reviewers for their thoughtful comments and efforts towards improving our manuscript. In the following, we address comments specific to each reviewer:

Reviewer 1

In Introduction, please delete the first paragraph containing the recommendations from the editorial office. “The introduction should briefly place the study in a broad context and highlight why it is important. It should define the purpose of the work and its significance. The current state of the research field should be carefully reviewed and key publications cited. Please highlight controversial and diverging hypotheses when necessary. Finally, briefly mention the main aim of the work and highlight the principal conclusions. As far as possible, please keep the introduction comprehensible to scientists outside your particular field of research. References should be numbered in order of appearance and indicated by a numeral or numerals in square brackets—e.g., [1] or [2,3], or [4–6]. See the end of the document for further details on references.”

Thank you for noticing this formatting mistake. This paragraph was deleted from the manuscript.